# A Configurable Anonymisation Service for Semantically Annotated Data

Paul Feichtenschlager[1,*], Christoph Fabianek[1], Fajar J. Ekaputra[2], Sebastian Haas[3] and Gabriel Unterholzer[1]

[1]*OwnYourData.eu, Michael Scherz-Straße 14, 2540 Bad Vöslau, Austria*

[2]*Vienna University of Economics and Business, Welthandeltsplatz 1, 1020 Vienna, Austria*

[3]*Hinterland Systems, Hörlgasse 10, 1090 Vienna, Austria*

### Abstract

Due to the increasing push for data sharing in Renewable Energy Communities (RECs)—driven by evolving regulations and energy transition goals—there is a growing need for trusted, privacy-preserving data-sharing infrastructures. One of the key challenges for RECs is enabling internal and external data sharing while minimizing privacy risks and preserving data value. In this work, we introduce a generic, configurable online anonymisation service for semantically annotated data as an extension to the Semantic Overlay Architecture (SOyA). Our approach employs an automated rule-based anonymisation pipeline that reduce re-identification risks while maintaining the utility of shared data. We demonstrated how this service can support compliant, secure, and practical data-sharing practices in real-world REC scenarios.

### Keywords

Data anonymisation, Renewable Energy Communities, Governance, Semantic annotation

## 1. Introduction

The digital transformation of the energy sector, driven by the increasing deployment of decentralized energy resources and smart grid technologies, has brought both opportunities and challenges. Renewable Energy Communities (RECs) are at the forefront of this transformation, enabling citizens to jointly produce, consume, and manage renewable energy. A critical technical enabler of these communities is the use of smart meters, which capture high-resolution energy data at the household level.

In Austria, the rollout of smart metering systems is mandated by the Intelligente Messgeräte-Verordnung (IME-VO), which requires that at least 95% of households be equipped with smart meters by the end of 2024 [1]. These devices facilitate bidirectional communication and provide granular consumption and production data, often at 15-minute intervals. Furthermore, under §84 of the Elektrizitätswirtschafts- und -organisationsgesetz (ElWOG), network operators in Austria are obliged to collect, store, and make this data accessible to end-users via online portals—daily values by default, quarter-hourly data only upon explicit consent [2]. While this infrastructure supports greater energy efficiency and citizen participation, it also raises pressing concerns regarding privacy, data protection, and access control. Smart meter data, though seemingly technical, qualifies as personal data under the General Data Protection Regulation (GDPR) [3] due to its potential to reveal sensitive behavioural patterns, such as presence at home, appliance usage, or the PV output of a prosumer installation.

Against all this backdrop, one of the core challenge for the Renewable Energy Communities (RECs) is to provide a privacy-preserving mechanisms for sharing their data within the energy communities. Since energy consumption and supply data constitute sensitive personal information, linking such data to identifiable individuals must be strictly prevented. At the same time, these data can offer

*NeXt-generation Data Governance workshop 2025, September 03, 2025, Vienna, AT*

*Corresponding author.

✉ paul@ownyourdata.eu (P. Feichtenschlager); christoph@ownyourdata.eu (C. Fabianek); fajar.ekaputra@wu.ac.at (F. J. Ekaputra); seb@hinterland.systems (S. Haas); gabriel@ownyourdata.eu (G. Unterholzer)

 0009-0002-4410-8796 (C. Fabianek); 0000-0001-7116-9338 (F. J. Ekaputra)

valuable insights for analysis and decision-making. Therefore, a mechanism is needed that preserves the structural characteristics of the data while ensuring that individual identities cannot be inferred.

Currently, several tools and services are available for data anonymisation. For example the EU-funded open-source tool Amnesia supports techniques such as masking, k-anonymity, km-anonymity, and the computation of demographic statistics [4]. Geninvo provides a tailored solution for the secure sharing of clinical data [5]. The ARX Data Anonymisation Tool offers a wide range of anonymisation capabilities, including generalization, suppression, and risk analysis [6]. However, to the best of our knowledge, none of these tools offer an open, online service that enables the annotation of data with a layer that explicitly defines and enforces its anonymisation requirements.

This paper introduces a practical solution to one of the key open challenges in this regulatory and technical landscape: a privacy-preserving sharing of high-resolution PV production data within energy communities. We present the design and implementation of a software plugin for a DGA-compliant data intermediary, which performs automated anonymisation of REC data prior to any data sharing. Our contribution lies in demonstrating how automated, rule-based anonymisation pipelines can reduce re-identification risks while maintaining data utility, thus enabling compliant and trustworthy data sharing in real-world Renewable Energy Communities (RECs) scenarios.

## 2. Background and Related Work

This section outlines the background and related work relevant to this work. First, the foundations of SOyA, as a core component of our service are explained. Afterwards, this sections gives an overview of the most prominent anonymisation techniques.

### 2.1. Semantic Overlay Architecture (SOyA)

SOyA (Semantic Overlay Architecture) is an open-source framework designed to facilitate the authoring, validation, and transformation of semantically annotated data models [7]. It addresses the need for flexible yet rigorous data modeling in decentralized and privacy-sensitive environments—such as those found in energy data ecosystems. SOyA introduces a modular architecture based on two primary concepts: Bases, which define the core structure of a dataset, and Overlays, which semantically enrich the base by specifying annotations, classifications, constraints, or transformations. These overlays enable the separation of concerns by allowing different semantic layers (e.g., validation, encoding, personal data classification) to be composed as needed.

SOyA components are usually authored in YAML for ease of use and are ultimately compiled into JSON-LD 1.1, allowing compatibility with Semantic Web technologies and Linked Data tools. Each SOyA object is uniquely identified via a Decentralized Resource Identifier (DRI)—a content-based hash that ensures immutability and referential transparency. This mechanism aligns with principles of data sovereignty and auditability, critical in regulated domains such as energy data exchange. SOyA has been integrated into Semantic Containers, a lightweight execution environment for data processing, enabling privacy-preserving, modular pipelines. For the use case of renewable energy communities, SOyA enables the explicit modeling of personal data attributes (e.g., energy usage, timestamps) and facilitates the annotation of privacy-relevant elements. This functionality ensures that data intermediaries can apply standardized anonymisation or minimization logic before external sharing, in line with GDPR and Data Governance Act requirements.

### 2.2. Data Anonymisation

As data volumes grow and data mining technologies become more prevalent, concerns regarding data protection have also intensified. Personal data, in particular, is highly sensitive and demands careful handling [8]. Anonymisation aims to protect the privacy of personal data while preserving the structural integrity and utility of the underlying data [9]. Fung et al. [10] identify three key objectives in the anonymisation process: adherence to privacy constraints, preservation of data utility, and retention of

data truthfulness. To achieve this, direct identifiers and quasi-identifiers - attributes that can be used to reidentify individuals alone or in combination - must be removed or transformed appropriately [11].

One of the primary threats that anonymisation techniques seek to address is the linking attack, in which two datasets are combined using shared quasi-identifiers to reveal sensitive information. For example, voter registration data can be linked to medical records using attributes such as ZIP code, age, and gender, potentially making individuals' medical histories identifiable [12]. To mitigate this risk, Samarati and Sweeney introduced the concept of k-anonymity, which requires that each record in a dataset be indistinguishable from at least $k - 1$ other records with respect to a set of quasi-identifiers.

**Data Anonymisation Techniques.** To address the challenges inherent in the anonymisation process, various approaches have been proposed in the literature. We introduce two of the most prominent techniques in the following.

- **Generalization** involves replacing specific attribute values with more abstract or less precise representations, thereby grouping multiple records under a generalized value [9]. This technique is applicable for attributes that exhibit hierarchical or multi-level structures—such as geographic locations or for numerical attributes, where values can be grouped into ranges or buckets [8].
- **Randomization** adds a random salt to all values of an attribute. It is suitable only for attributes with cardinal measurement scales, where value distances are defined [14].

**Data Governance.** The governance of personal data within the European Union is shaped by a comprehensive legal framework that seeks to balance data utility with fundamental rights to privacy and data protection. Central to this framework is the General Data Protection Regulation (GDPR) [3], which has been in force since 2018 and applies to all organizations processing personal data of EU citizens. The GDPR establishes key principles such as data minimization, purpose limitation, and accountability, and it grants data subjects enforceable rights including access, rectification, and erasure [15, 16].

An important mechanism under the GDPR is *anonymisation*, defined as the irreversible process of removing personal identifiers such that data subjects are no longer identifiable. Once data is anonymised in this manner, it falls outside the scope of the GDPR, thus permitting its use for secondary purposes such as research, analytics, or data sharing without additional legal constraints [17]. This makes anonymisation a crucial tool for organizations aiming to utilize data in a compliant and privacy-preserving manner [15]. Complementing the GDPR, the *Data Governance Act* (DGA) introduces further provisions to foster the secure, trustworthy reuse of data—particularly data held by public bodies, data shared for altruistic purposes, and data exchanged through neutral intermediaries. The DGA sets out rules for the registration, obligations, and neutrality of data intermediaries to ensure that data-sharing services operate transparently and independently. By establishing a governance infrastructure for data reuse, the DGA extends the regulatory landscape from individual data rights to ecosystem-level accountability and trust.

Together, the GDPR and DGA form a layered governance model that simultaneously protects individuals and facilitates responsible data innovation within the EU.

## 3. The Data Anonymiser Service

In this work, we developed a service that applies anonymisation techniques to datasets containing personal data. The primary objective was to create a generic and configurable solution that ensures compliance with the General Data Protection Regulation (GDPR). To reduce the entry barrier for users, the service is designed to be easily configurable through SOyA using a simple YAML file.

The degree of anonymisation dynamically adapts to the size of the dataset and the number of attributes to be anonymised. This ensures that a sufficient level of anonymisation is applied while preserving the overall structure and utility of the original data set as much as possible.

The anonymisation service is provided as a free-to-use, web-based portal[1], which comprises of a lightweight single-page application built with Rails offers a responsive graphical user interface (GUI) that abstracts all protocol details, so that domain experts without programming experience can execute the same workflows that advanced users may invoke programmatically through the OpenAPI endpoint[2]. The front-end and back-end components are packaged as Docker images and are continuously deployed on an OwnYourData Kubernetes cluster; nightly builds are published under an MIT licence to foster reproducibility and community contributions. The open-source code is available in the accompanying GitHub repositories[3].

To meet GDPR and DGA accountability requirements, every request is logged in an append-only audit ledger, while raw input files are stored only for the duration of processing and deleted immediately afterwards. Organisations with stricter data-sovereignty needs can deploy the service on-premises via the provided Docker images, achieving functional parity with the managed Software-as-a-Service instance but retaining full control over network boundaries and compliance monitoring. This dual provisioning model—public SaaS and self-hosted container images—minimises the entry barrier for exploratory usage while ensuring that production environments can satisfy sector-specific governance constraints.

## 3.1. Data Anonymisation Workflow

The service accepts requests containing both the data to be anonymised and a URL pointing to a valid anonymisation configuration. The process begins by fetching the configuration from the provided URL. Once validated, an anonymiser object is created for each attribute based on the configuration. This is enabled through an anonymiser interface, which is implemented by various anonymisation strategies described in Section 3.2.

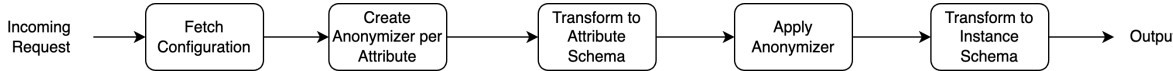

**Figure 1:** Workflow of anonymisation Service

To apply anonymisation, the input data is first converted into an attribute-oriented schema, and for each attribute, the service aggregates values across all instances, while allowing attributes to remain blank for instances where no value is provided. After the transformation, anonymisers are applied to the aggregated attribute values. The number of values per attribute remains unchanged throughout the process. If an instance lacks a value for a specific attribute, that attribute remains empty in the anonymised dataset. Finally, the anonymised attributes are transformed back to an instance oriented data structure, to produce the final output.

## 3.2. Data Anonymisation Implementation

The anonymisation service is designed to allow the seamless integration of custom anonymisation implementations. Each anonymiser must implement a service interface that accepts a list of attribute values along with the total attribute count, and returns a corresponding list of anonymised values. This work introduces two anonymiser implementations: (a) **Generalisation** and (b) **Randomisation**, each tailored to different data types and anonymisation strategies.

Both anonymisation techniques require the definition of a specific number of groups. To ensure a sufficient level of anonymity, the desired number of generalization groups $g$ is calculated based on the dataset size $k$ and the number of anonymised attributes $n$. The objective is to achieve a 99% probability that no instance in the dataset remains unique after anonymisation.

---

[1]https://anonymiser.ownyourdata.eu
[2]https://anonymizer.go-data.at
[3]https://github.com/OwnYourData/anonymisation-service

To determine the number of groups under this requirement, an approach by Jiang et al. [18] to quantify re-identification risk, was adapted. Assuming that the attributes in the input data are statistically independent and that all groups are of equal size, a formula was derived to guarantee that the probability of any individual being uniquely identifiable is less than 1%. Given $k$ as the number of individuals and $n$ as the number of anonymised attributes, the number of required groups $g$ is computed as:

$$g = \frac{1}{\sqrt[n]{1 - \sqrt[k]{1 - \sqrt[k]{0.99}}}} \tag{1}$$

**Generalization.** In generalization, attribute values are grouped into categories, and the corresponding group labels are written in the anonymised dataset. The number of groups $g$ is defined in Equation 1.

Values are assigned to buckets in such a way that each group contains the same number of values. Outliers are not assigned to separate groups, avoiding easy re-identification. Instances are first sorted by value and then assigned to groups on the basis of their position. Each group is then labeled on the basis of the values it contains.

Generalization also supports object-type attributes, assuming a hierarchical structure is defined in the configuration. The algorithm traverses the hierarchy from the most specific level upward, reducing each data point to a generalized value. A valid generalization is accepted when:

- The number of resulting groups is less than or equal to the calculated number of groups $g$.
- Each group contains at least $\frac{k}{2g}$ elements, where $k$ is the size of the dataset and $g$ is the calculated number of groups.

These two requirements ensure that the resulting generalization maintains the desired level of abstraction and prevents the formation of groups that consist only of outliers, which would otherwise be easily identifiable.

**Randomization.** In the randomization process, noise is added to attribute values based on a normal distribution. The intensity of this noise depends on the distribution of the dataset: data points in sparse regions receive increased noise to enhance their anonymity.

The parameter $i$ represents the number of instances that would be assigned to a bucket if generalization were applied. It is calculated by dividing the dataset size $k$ by the number of buckets $g$. The salt applied to a value $x$ is then computed by multiplying a standard normal random variable with zero mean and unit variance $\mathcal{N}(0, 1)$, by the distance between $x$ and its $i$-th closest value $x_i$. The formula is defined as follows:

$$salt_x = \mathcal{N}(0, 1) * \text{distance}(x, x_i) \tag{2}$$

To generate an anonymised value, the salt is added to the original value. If the resulting value falls outside the range of values of the original dataset, the salt is instead subtracted from the original value.

**Running Example.** Figure 2 illustrates an anonymisation example with the data of an REC. The input dataset contains the attributes entryDate, longitude, latitude, and energyProduction. anonymisation is applied to the first three attributes, while energy production remains unchanged. As a result, the energy data is no longer linkable to a specific individual.

The anonymisation service is invoked with two primary components: the input data to be anonymised and a reference URL pointing to the anonymisation configuration. The configuration defines the anonymisation methods applied to each attribute. Despite the anonymisation, data mining remains feasible; for example, the dataset can still be used to analyze energy production patterns across different geographic locations. A sample dataset and detailed instructions for reproducing the example are available in the public GitHub repository.

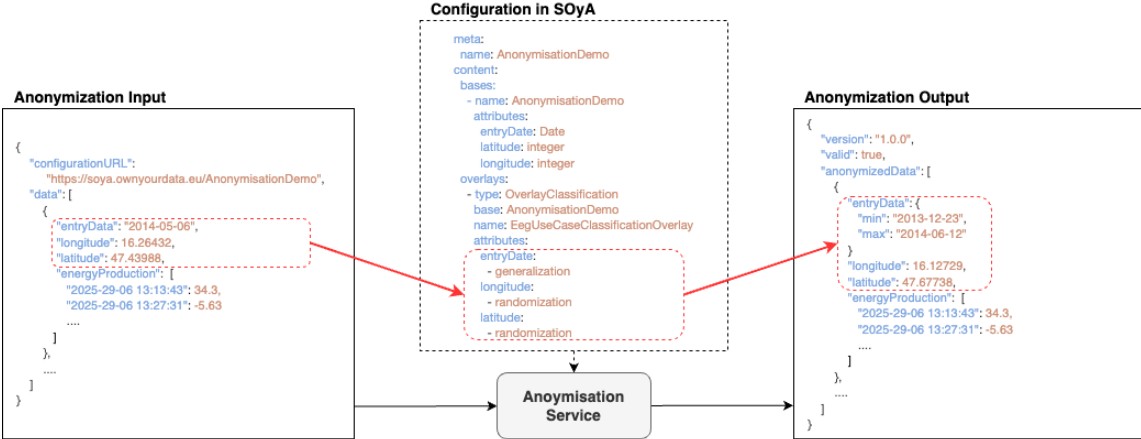

**Figure 2:** An example anonymisation on REC data

# 4. Feasibility Evaluation

To assess the data security of the service, a series of tests was conducted using synthetic test datasets. These datasets were generated to replicate the personal data of members of a Renewable Energy Communities (RECs) in the Austrian state Burgenland. Both generalization and randomization techniques were applied to these datasets, with varying sample sizes ranging from 100 to 10.000. This section explains the synthetic test data generation process and outlines the method used to calculate a benchmark value. Additionally, we present the test results and evaluate the compliance with existing regulatory standards.

**Test Data Generation.** Due to the limited availability of real personal data, synthetic data was generated. The created data sets include three attributes: longitude, latitude, and the date of an individual's registration in the energy community. The geographic coordinates were selected to approximate locations within the Austrian state of Burgenland. Registration dates were randomly generated within the range from 2005 to 2025, with a higher probability assigned to more recent years, reflecting the assumption that membership registrations have increased in recent years.

**Benchmark Calculation.** To evaluate the anonymisation service with respect to data confidentiality, k-anonymity is employed as the metric for generalization-based techniques. For randomization-based anonymisation, a modified approach is required, that assesses the similarity between original and anonymised instances to establish an analogous notion of k-anonymity.

Similarity in this context is defined as follows: an original instance $A_{original}$ is considered similar to an anonymised instance $B_{anonymised}$ if $A \neq B$ and all randomized attribute values in $B_{anonymised}$ lie within an acceptable range of deviation from the corresponding values in $A_{original}$. To determine this acceptable range, the distribution of distances between original values and their anonymised counterparts is analyzed. Since the randomization process adds normally distributed noise (salt), a threshold of $2\sigma$ (twice the standard deviation) is used as the similarity criterion. If the distance between an anonymised value and the corresponding original value is below this threshold, the two are considered similar for that attribute.

An anonymised instance is considered similar to an original instance if it satisfies the similarity condition for all attributes. The k-anonymity of the anonymised dataset is then defined as the minimum number of original instances that are similar to any given anonymised instance.

**Evaluation Result.** The test results are presented in Table 1. Both randomization and generalization were evaluated using sample sizes of 100, 1.000, and 10.000. Each configuration was executed ten times with different synthetic datasets. For each setup, the median and minimum k-anonymity values across

the ten runs are reported. In addition, the table includes the number of buckets (groups) used for each sample size, calculated according to the method described in Section 3.2.

| | Size: 100 Groups: 2 | | Size: 1.000 Groups: 4 | | Size: 10.000 Groups: 8 | |
|---|---|---|---|---|---|---|
| | Median | Minimum | Median | Minimum | Median | Minimum |
| **Generalization** | 9.0 | 6 | 7.0 | 5 | 7.0 | 6 |
| **Randomization** | 24.5 | 18 | 17.5 | 8 | 16.5 | 12 |

**Table 1**
Median and minimum k-anonymity for Generalization and Randomization across different dataset sizes

Our experiment demonstrates that the data anonymised by our service comply with GDPR requirements. Across all tested sample sizes, no instance could be uniquely identified based on its anonymised attributes, regardless of of the chosen anonymisation techniques. In all 60 tests conducted, each anonymised instance was indistinguishable from at least five others, satisfying a minimum k-anonymity level of 5. However, the analytical utility of the anonymised datasets varies depending on the data size and in further consequence the number of groups employed. Figure 4 shows the spatial grouping by data size when generalization is used, e.g., a sample size of 100 results in only four spatial groups (i.e., two group of latitudes and longitudes), limiting the potential for meaningful analysis.

Overall, randomization achieved a higher degree of anonymisation than generalization. Nevertheless, the selected group count appears to offer a reasonable trade-off between data confidentiality and analytical utility for both approaches. For scenarios demanding stronger privacy guarantees, the number of groups can be reduced at the cost of analytical detail.

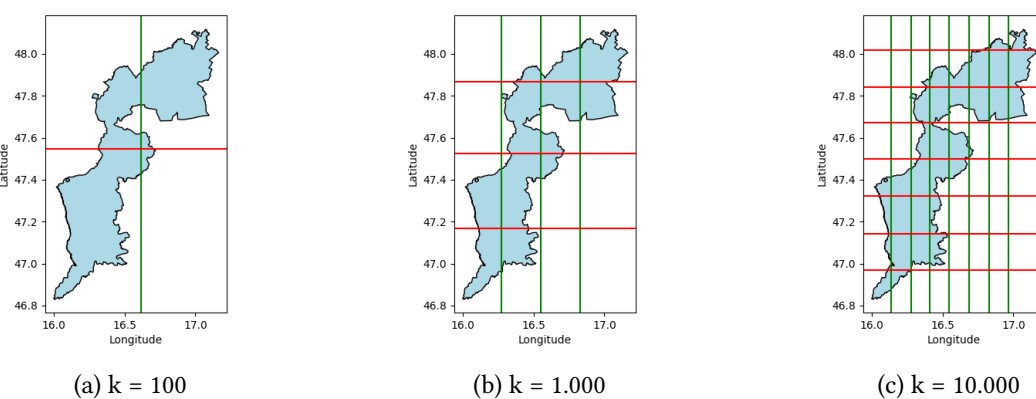

(a) k = 100         (b) k = 1.000         (c) k = 10.000

**Figure 3:** Spatial Groups according to Data size

# 5. Summary and Future Work

This work presents a configurable and extensible anonymisation service featuring an open interface, tailored to semantically annotated datasets, focused on GDPR compliance and demonstrated through an initial feasibility study in the context of Renewable Energy Communities. The service supports generalization and randomization techniques, integrates with SOyA for declarative configuration, and is accessible via both a public API and a user-friendly web interface. Future work will focus on extending the service to further anonymisation techniques. Enhancing the expressiveness of anonymisation outputs is another key objective: future releases will compute relevant KPIs to provide better insight into the characteristics of the anonymised datasets. Lastly, the user interface will be iteratively improved based on systematic user feedback to increase accessibility and usability for non-technical domain experts.

## Acknowledgments

This work was conducted as part of the USEFLEDS project. This project has received funding in the program "Datenökosysteme für die Energiewende" by the Federal Ministry for Climate Action, Environment, Energy, Mobility, Innovation and Technology (BMK) under grant number 905128.

## Declaration on Generative AI

During the preparation of this work, the author(s) used X-GPT-4 in order to: Grammar and spelling check. After using these tool(s)/service(s), the author(s) reviewed and edited the content as needed and take(s) full responsibility for the publication's content.

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
