# OpenReview forum: "A Configurable Anonymisation Service for Semantically Annotated Data"
_SEMANTiCS.cc/2025/Workshop/NXDG — NXDG 2025 Conditionallyed_

### Official Review · ~Ben_De_Meester1 · 2025-07-15
**Although certainly effective and deemed useful, the paper fails to give a good argumentation and fitting evaluation to make it relevant for the workshop's topics.**

**Rating:** 4
**Confidence:** 4

**Review:**

### High-level remarks

- A lot of introduction space is spent on the context, but the need for "an open, online service that enables the annotation of data with a layer that explicitly defines and enforces its anonymisation requirements" isn't properly introduced. Yes anonymization is needed, but it's unclear why this must happen using an online service, and why anonymisation requirements must be explicitly annotated --> why not hardcode this is the energy intermediary? Is it a good idea to make this configurable (by whom)? Shouldn't the anonymization rules be predefined by the sector and mandated?
- "We present the design and implementation of a software plugin for a DGA-compliant data intermediary, which performs automated anonymisation of REC data prior to any data sharing." --> I really try, but I cannot find any mentioning of a plugin or any relation to data sharing ór DGA ór data intermediary in the remainder of the paper. Without this further mentioning, the topic does not seem very fitting for the workshop - it's a bit related to 'user-centric perspectives on privacy and data protection', however, the actual contribution (as evidenced by the evaluation and correctly reflected in the title) is "A Configurable Anonymisation Service for Semantically Annotated Data".
  - To make it more fitting with the workshop, I would expect, e.g. a discussion on what the impact would be on a data intermediary, e.g. by checking performance and how much processing it requires
- How is this "semantically annotated" data? The SOyA all seem to be focussed on JSON (so sure, JSONLD as byproduct is _something_), however, the demonstrator really focuses on plain JSON (i.e. no @context is provided), the TTL export does not cover the same info (e.g. no Latitude min/max info is provided), and none of the minted predicate and class links are resolvable (e.g. https://soya.ownyourdata.eu/AnonymisationDemo/Geburtsdatum gives a 404). So yes, you provide an RDF export, but the actual data is not semantically annotated.
- the link https://github.com/OwnYourData/anonymisation-service gives a 404. this MUST be fixed if the paper would be published.
- given the main body of text is within the 7 page limit, I assume this to be a short paper (I assume Acknowledgements and Declaration on Generative AI are not expected to be part of the body text). My scoring is given for a short paper.

### Detail comments

#### Introduction

- "one of the key open challenges in this regulatory and technical landscape" --> is it? how do you know? do you claim this is key, or do you have a reference?

#### The Data Anonymiser Service

- the link https://github.com/OwnYourData/anonymisation-service gives a 404. I tried to look into https://github.com/OwnYourData/anonymiser but I couldn't any actual code, it's mostly an API wrapper around another API, for which I couldn't find the source.

#### Feasibility Evaluation

- Evaluation set-up seems well done
- Figure 3 is not mentioned in the text.

#### References

- Reference [5] is incomplete

---

### Official Review · ~Ross_Horne1 · 2025-07-23
**REC**

**Rating:** 6
**Confidence:** 3

**Review:**

The paper addresses the privacy issue of re-identification while sharing data. This is in the context of data sharing in the renewable energy industry.

k-anonymity is considered. There are however many known re-identification attacks on k-anonymity that could be mentioned at acknowledged. This are is notorious for there being no fool-proof solution. The work goes on to evaluate an implementation of a system offering some anonymity.

The paper is missing key context such as the novelty of their system compared to others implementing such a solution for anonymity. This could be achieved through a combination of further citations and more explicit statement on novelty and contribution in the conclusion.

As a working demo of something personal data on the Web related, this is however within scope of what can be presented in this venue.

---

### Decision · Program_Chairs · 2025-07-25

Conditionally Accepted